# Evidence of Copper Nanoparticles and Poly I:C Modulating Cas9 Interaction and Cleavage of COR (Conserved Omicron RNA)

**DOI:** 10.3390/bioengineering10050512

**Published:** 2023-04-25

**Authors:** Lindy G. Karrer, Elza Neelima Mathew, Juliet Nava-Chavez, Abeera Bhatti, Robert K. Delong

**Affiliations:** 1Division of Biology, Kansas State University, Manhattan, KS 66506, USA; karrer@ksu.edu; 2Department of Anatomy and Physiology, Kansas State University, Manhattan, KS 66506, USA; julietn@ksu.edu (J.N.-C.); abeerabhatti1@gmail.com (A.B.); 3Landmark Bio, Innovation Development Laboratory, Watertown, MA 02472, USA; rdelong@landmarkbio.com

**Keywords:** Cas9, nanoparticles, copper, RNA, SARS-CoV-2, omicron, poly I:C

## Abstract

Conserved omicron RNA (COR) is a 40 base long 99.9% conserved sequence in SARS-CoV-2 Omicron variant, predicted to form a stable stem loop, the targeted cleavage of which can be an ideal next step in controlling the spread of variants. The Cas9 enzyme has been traditionally utilized for gene editing and DNA cleavage. Previously Cas9 has been shown to be capable of RNA editing under certain conditions. Here we investigated the ability of Cas9 to bind to single-stranded conserved omicron RNA (COR) and examined the effect of copper nanoparticles (Cu NPs) and/or polyinosinic-polycytidilic acid (poly I:C) on the RNA cleavage ability of Cas9. The interaction of the Cas9 enzyme and COR with Cu NPs was shown by dynamic light scattering (DLS) and zeta potential measurements and was confirmed by two-dimensional fluorescence difference spectroscopy (2-D FDS). The interaction with and enhanced cleavage of COR by Cas9 in the presence of Cu NPs and poly I:C was shown by agarose gel electrophoresis. These data suggest that Cas9-mediated RNA cleavage may be potentiated at the nanoscale level in the presence of nanoparticles and a secondary RNA component. Further explorations *in vitro* and *in vivo* may contribute to the development of a better cellular delivery platform for Cas9.

## 1. Introduction

The emergence of variants of SARS-CoV-2 began concomitantly with the COVID-19 pandemic. In 2021, Omicron was classified as a variant of concern (VOC) by the SARS-CoV-2 Interagency Group (SIG), USA [1]. Variants keep emerging due to genetic modification amidst the use of vaccines and antiviral therapeutic regimes. Genetic modifications are mostly noted in the spike protein, which is responsible for membrane fusion and viral entry into cells. Non-spike mutations are being observed in novel variants [1,2]. Recombinant spike protein or nucleoside-modified mRNA encoding the spike protein or vector encoding a stabilized variant of the protein is the major ingredient of the currently available vaccines against the disease [3]. Using an RNA genome-wide analysis, it was discovered that conserved omicron RNA (COR), a 40 base long 99.9% conserved sequence in SARS-CoV-2 Omicron variant, predicted to form a stable stem loop, the targeted cleavage of which can effectively curb the rapid spread of variants [4]. Exploration of the possibility of inactivating COR thus becomes an alluring path worth diligently pursuing in search of better treatment, prevention, and control of SARS-CoV-2.

The clustered regularly interspaced short palindromic repeats (CRISPR)-CRISPR-associated proteins (Cas) is a system of microbial defense against viruses first discovered in *Escherichia coli* [5,6,7]. The advent of cleaving double-stranded DNA using RNA-guided nucleases called CRISPR-associated proteins (Cas) has revolutionized the field of gene editing. Briefly, Cas is guided to a specific target by a single guide RNA (sgRNA) which consists of two portions—a target-specific crRNA sequence and a tracrRNA sequence interacting with Cas. This target-specific binding is dependent on the protospacer-adjacent motif (PAM) in the target RNA [8,9]. Depending on the type of Cas protein involved, there are numerous types of CRISPR-Cas systems. Three widely recognized major systems are, (1) Type I with Cas3 (2) Type II with Cas9, and (3) Type III with Cas10 [7,10]. Of these, the type II system using Cas9 as its effector protein is the best understood [11,12].

In addition to DNA cleavage, there is evidence that Cas9 can also cleave single-stranded RNA in a PAM-dependent or -independent manner [13,14]. RNA-based RNA cleavage enables a unique RNA silencing system in prokaryotes [15]. Utilizing this system, RNA viral genes have been knocked down in PRRS viral genomes, which can be extended to search for potent antiviral therapeutics [16]. Even though the cellular delivery of Cas9 and sgRNA is easily feasible in a laboratory, it faces many challenges when it comes to animal or human patients. Common laboratory methods for transfection are not applicable in live patients. In addition, Cas9 has a high molecular weight of ~160 kDa and the sgRNA has a phosphate backbone that is negatively charged. In the search for better delivery methods, different physical and chemical approaches are in use. The best delivery platform delivers Cas9 to target tissues ensuring the passage of Cas9 across the cell membrane, overcoming cellular degradation pathways [17,18].

Nanoparticles, organic, inorganic, metallic, or polymeric, are considered a good tool in developing drug delivery systems for diagnosis as well as therapy [19]. Original or surface-functionalized metal-based nanoparticles are also effective antiviral agents. It was discovered that the antiviral activity of multiple nanoparticles is related to their protein interaction, inhibition, or activation of certain enzymes involved in virus infection, replication, or life cycle [20,21]. Previously, we observed that copper or zinc oxide nanoparticles affect the “star activity” of DNA restriction endonucleases altering their sequence-specific DNA cleavage [unpublished data]. Nanoparticles can denature viral glycoproteins by surface attachment and disulfide bond disruption. The release of metal ions from nanoparticles, the production of reactive oxygen species, damage to microbial nucleic acid/protein, and the loss of cellular integrity are regarded as the major mechanisms behind this antimicrobial activity [22,23,24]. Copper nanoparticles (Cu NPs) have been shown to induce contact killing of the influenza A virus, which was attributed to the disruption of the viral genome possibly due to ROS generation [25]. Surface-related catalytic activity might also contribute to the SARS-CoV-2 antiviral activity of copper nanoparticles [26]. Painting surfaces with Cu NPs fabricated from cuprous oxide (Cu_2_O) was found to inactivate >99% SARS-CoV-2 infectivity, suggesting that Cu NP painting in public areas is recommended [27].

Polyinosinic-polycytidilic acid (poly I:C) is a double-stranded synthetic viral mimetic RNA with immunogenic properties [28]. The immunogenic activity of poly I:C, attributed to its ability to activate Toll-like receptor 2 (TLR3) and melanoma differentiation-associated protein5 (MDA5), makes it an antiviral as well as an antitumoral agent [29,30,31]. Here, the possibility of using poly I:C as a wobble guide RNA for Cas9 to bind to COR is explored.

The key theme of this study was to investigate whether Cas9 is able to cleave COR and whether that interaction is affected by the presence of poly I:C and Cu NPs. This multi-component approach is expected to make use of the antiviral activities of each entity against SARS-CoV-2. To define the intracellular behavior of this approach, characterization, and understanding of the biophysical interactions among the different entities is required [32].

## 2. Materials and Methods

Copper nanoparticles (Cu NPs; size 13–40 nm; dry powder) were obtained from PlasmaChem GmbH (Berlin, Germany). Cas9 nuclease GFP NLS protein was purchased from Applied Biological Materials Inc. (Richmond, BC, Canada) and stored at −20 °C. Conserved Omicron RNA (COR) [5′-rCrUrU rCrUrG rCrUrG rCrUrC rUrUrC rArArC rCrUrG rArArG rArArG rArGrC rArArG rArArG rArArG rA-3′] was custom synthesized by Integrated DNA Technologies (Coralville, IA, USA) and stored at −20 °C [4]. Polyinosinic-polycytidylic acid sodium salt (poly I:C) was obtained from Sigma Aldrich (St. Louis, MO, USA) and stored at −20 °C. RNase-free water was used in the preparation of all solutions unless specified otherwise. RNase-free water was prepared by treating with diethylpyrocarbonate (DEPC) (Sigma Aldrich, St. Louis, MO, USA) at the rate of 1 mL DEPC per liter of water (0.1% DEPC), stirring for 2 h followed by autoclaving. Cu NP solution (1 mg/mL) was prepared in RNase-free water by vortex mixing and stored at 4 °C.For the biophysical characterization of interactions between Cu NPs, Cas9, poly I:C, and COR, the following techniques were used: (1) Dynamic Light Scattering (DLS) and Zeta Potential Measurements, (2) Agarose Gel Electrophoresis, and (3) Two-Dimensional Fluorescence Difference Spectroscopy (2-D FDS) [30,31,32,33,34].

Dynamic light scattering and zeta potential measurements were performed using a Malvern Zetasizer Nano ZSP (Worcestershire, UK). Agarose gel electrophoresis was done using the Owl™ EasyCast™ B2 Mini Gel Electrophoresis Systems (Thermo Scientific, Waltham, MA, USA), and gel imaging was conducted using a Bio-Rad Gel Doc^TM^ XR+ gel documentation system (Hercules, CA, USA). Fluorescence measurements were obtained using a Spectramax^®^ i3x multi-mode microplate reader (Molecular Devices (San Jose, CA, USA)).

### 2.1. Dynamic Light Scattering (DLS) and Zeta Potential Measurements

Dynamic light scattering (DLS) to determine the particle size and polydispersity indices, and zeta potential measurements were performed for Cu NPs, Cu NPs + Cas9, Cu NPs + COR, Cu NPs + poly I:C, Cu NPs + COR+Cas9, Cu NPs + poly I:C + COR + Cas9, and Cu NPs + COR + poly I:C + Cas9. The last two treatments differ in the order of addition of poly I:C and COR. All the samples contained 20 µg of Cu NPs. All samples were incubated for 15 min at 150 rpm on an orbital shaker (MIDSCI, Fenton, MO, USA) and were dispersed in RNase-free water and loaded into folded capillary cell cuvettes (Malvern, Worcestershire, UK) followed by reading in the Malvern Zetasizer Nano Range Dynamic Light Scattering (DLS) equipment (Malvern, Worcestershire, UK)

The composition of each sample treatment is shown in Table 1. RNase-free water was used to adjust the volume to 1 mL. For each treatment, two separate trials were conducted. Each trial assessed the properties of two different tubes. Dynamic light scattering (DLS) was used to determine the particle size and polydispersity indices, and zeta potential measurements were conducted from the same set of tubes.

### 2.2. Agarose Gel Electrophoresis

Agarose gel electrophoresis was performed to visualize the effect of the interaction of Cu NPs, Cas9, and poly I:C on the quality and quantity of COR. Cu NP stock of 1 mg/mL was diluted down to 1:200 using Milli-Q water (Millipore Sigma, Burlington, VT, USA). Poly I:C stock of 1 mg/mL concentration was diluted to 1:100 using Milli-Q water. COR stock consisted of 2 µL COR diluted in 18 µL Milli-Q water. Samples were prepared as shown in Table 2 and incubated at 37 °C for 15 min.

After incubation, 10 µL of saturated sucrose and 1 µL of Bullseye DNA SafeStain (MIDSCI, Fenton, MO, USA) were added to each sample. A total of 20 µL of each sample was loaded onto a 2% agarose gel and ran at 100 V for 30 min in the gel electrophoresis system followed by imaging.

### 2.3. Two-Dimensional Fluorescence Difference Spectroscopy (2-D FDS)

Two-Dimensional Fluorescence Difference Spectroscopy (2-D FDS) was performed to detect changes in the fluorescence spectrum brought about by the interaction of Cu NPs, Cas9, poly I:C, and COR. Working stock of poly I:C (1 mg/mL), Cu NPs (1 mg/mL), Cas9 (4 mg/mL), and COR (0.1 units in 100 µL) were prepared in RNase-free water. 15 µL of COR from the stock solution was mixed with 10 µL of Hoechst Dye (Molecular Probes, Eugene, OR, USA) using a Vortex mixer (MIDSCI, Fenton, MO, USA). A total of 4 µL of the COR-Hoechst dye mix was added to all the samples. Fluorescence spectra were measured for the COR-dye mixture, Cu NPs, Cu NPs + Cas9, and Cu NPs + Cas9 + poly I:C using the optimization wizard function in the Spectramax i3x multi-mode microplate reader. The compositions of the different sample treatments were as shown in Table 3.

## 3. Results and Discussion

### 3.1. Zeta Potential and Particle Size Changes as Cas9 and COR Interact with Copper Nanoparticles

The Zeta potential of a nanoparticle is the electrostatic potential between the ionic double layer close to its surface and is related to the charge density at the particle surface [33,35]. Interaction with other materials or biomolecules affects the zeta potential, the change in which can be used as a quantitative measure of such interactions. This measurement is an indicator of the overall charge and stability of nanomaterials and their conjugates with biomolecules. Measurement values ranging from −30 mV to +30 mV are considered stable with less tendency for agglomeration. Physicochemical properties of the solution such as pH, ionic strength, nature of the surface ligands, temperature, and radiation can affect the zeta potential readings [36].

Following incubation with Cas9, COR, or poly I:C at room temperature, the zeta potential measurements of Cu NPs were examined to analyze the interaction between the species. The results are shown in Figure 1.

Zeta potential is expressed in millivolts (mV). All the readings are greaterthan −30 mV, which indicates the stability of the preparations. Consistent with our experience and preliminary data, zeta potential data for copper nanoparticle interaction with protein is inherently variable. The zeta potential of Cu NPs alone was −16.7 mV. Following incubation with Cas9, COR, poly I:C, and their combinations, the zeta potential of the Cu NPs showed a positive shift trend but was not significantly different from control. Except for Cas9 alone, all other additions of Cas9, COR, and/or poly I:C resulted in zeta potentials in the neutral range of −10 to +10 mV. A positive shift in zeta potential was expected when highly cationic Cas9 protein (charge: +22) interacts with Cu NPs [37]. However, the positive shifts with COR and poly I:C were surprising, as negative shifts were expected with both entities being negatively charged nucleic acids. The use of RNase-free water, necessary for these RNA experiments but which may not be deionized, may have complicated these results.

Changes in size and polydispersity indices were also monitored across the same samples (Table 4). The results shown are the average of the results from two separate trials, each with two different tubes of treatments.

Dynamic light scattering determines the size and size distribution of nanoparticles dispersed in a colloidal system. This technique measures the fluctuations in the intensity of light scattered from particles due to Brownian motion [38,39]. Size is expressed as diameter values in nanometers (nm). The size of the Cu NPs according to the DLS was 394 ± 978 d. nm. A clear increase in the size was observed with the addition of COR (6005 ± 90 d. nm) or Cas9 (5513 ± 246 d. nm) to the Cu NPs. When both COR and Cas9 were added, the increase was even more obvious (727 ± 2754 d. nm). However, with the addition of poly I:C, the size values observed were lower (3149 ± 1468) than the initial size of the Cu NPs. Even though the addition of Cas9, COR, and poly I:C increased the size value irrespective of their order of addition, these values were lower than those observed with the addition of Cas9 or COR alone.

The polydispersity index is a measure of uniformity of size distribution in a solution, with values ranging from 0 for homogenous or monodisperse solutions to 1 for highly heterogenous or polydisperse solutions. This attribute is dimensionless. Values less than 0.05 are more common in monodisperse samples. Values more than 0.7 are indicative of a very broad size distribution within the sample [40,41]. All the samples had polydispersity indices less than 0.7 as shown in Table 4.

### 3.2. Degradation of COR upon Cas9 Binding Possibly Enhanced by CuNPs and poly I:C

Agarose gel electrophoresis is a simple and inexpensive common tool used to visualize the presence and stability of nucleic acids. This method was used to visualize the stability of DNA and RNA (poly I:C) in the presence of zinc oxide and magnesium oxide nanoparticles [42]. The binding of gold nanoparticles to plasmid DNA has been confirmed by band shifts upon gel electrophoresis [43]. An interaction with nanoparticles changes the electrophoretic mobility of biomolecules such as protein and nucleic acid, which is visualized on the gel as a band shift [44]. Agarose gel electrophoresis is used to separate colloidal nanomaterials based on their size and shape, DNA-conjugated nanoparticles based on the number of DNA ligands per particle, and to identify and isolate the desired product from a large assembly of DNA-assembled nanomaterials [45,46]. Here, the technique was used to examine the effect of Cu NPs and poly I:C on the COR and COR–Cas9 complex.

The results of agarose gel electrophoresis are shown in Figure 2. Bands are counted from left to right. Band 1 shows COR alone and band 2 shows the COR-Cas9 complex. It appears that a major portion of COR was retained in the sample well itself when it was bound to Cas9. Retention was not observed when Cu NPs were introduced into the mix. However, the COR band becomes less intense (band 3). The band intensity is further reduced in the presence of poly I:C as is evident with band 4 (Figure 2). These reductions in band intensities along with the slight band shifts may be attributed to the degradation of COR upon Cas9 binding, which is enhanced in the presence of Cu NPs and poly I:C.

Agarose gel electrophoresis is not just a nucleotide and/or nanomaterial visualization tool but also helps to ensure the integrity of the material. Green synthesized Cu NPs (3.75 mM) have been used to isolate intact good-quality DNA from minimal quantities of blood and skeletal remains, which was clearly visualized using agarose gel electrophoresis [47].

### 3.3. Shifts in the Fluorescence Spectra from Two-Dimensional Fluorescence Difference Spectroscopy (2-D FDS)

Two-Dimensional Fluorescence Difference Spectroscopy (2-D FDS) is a technique that detects shifts in the fluorescence spectra of nanoparticles upon binding with biomolecules such as peptides, proteins, and nucleotides. A fluorescence spectral signature encompasses the excitation and emission wavelengths as well as the fluorescence intensity [48,49,50]. Using 2-D FDS, the spectral signatures of COR, COR + Cu NPs, COR + Cu NPs + Cas9, and COR + Cu NPs+ Cas9 + poly I:C were compared and the spectral signatures are shown in Figure 3. Excitation and emission spectra are expressed in nanometers (nm) and fluorescence intensity is expressed in relative fluorescence units (RFU). A difference in fluorescence excitation, emission, and RFU was seen when COR interacts with Cu NPs with slight quenching of the signal from a fluorescence intensity of 4.5 k to 4 k. Interestingly, when the Cas9 was added to the system, given its GFP labeling, the excitation maximum shifted consistent with that but the emission was of a longer wavelength with higher fluorescent yield (12.5 k), further supporting an interaction between the protein and nanoparticle. What was very insightful mechanistically, is the difference in the poly I:C data. The COR molecule being similar in size to a guide RNA sequence suggests that the 2-D FDS technique is RNA-dependent for Cas9. The addition of the poly I:C greatly quenches the fluorescence seen with COR, suggesting that there is some competition and loss of the COR from the complex. Indeed, the data look very similar to the COR alone, likely reflecting the interaction now between the COR:poly I:C, hence the reason for the change in cleavage pattern as seen in the gel.

## 4. Conclusions

Even with vaccination and antiviral treatment regimes in place, variants of SARS-CoV-2 keep emerging due to genetic mutation. This drives the need for exploring novel as well as existing avenues for treatment, prevention, and control of this viral disease. Conserved omicron mRNA (COR) is a 40 baselong 99% conserved sequence in SARS-CoV-2 Omicron variant, predicted to form a stable stem loop, the targeted cleavage of which will be an ideal next step in controlling the spread of variants. Mitigating the activity of Cas9 to RNA cleavage mediated by nanoparticle biomolecular interaction is one novel approach toward this overall goal, as described here.

CRISPR-Cas9, a microbial defense system against viruses, has been at the center stage of gene editing tools recently [5,6,7]. Cas9 is capable of editing RNA as well as DNA [13,14,15,16]. The development of the best intracellular delivery vehicle for Cas9 is challenging due to the large size of the protein and several other factors [17,18]. This study analyzed the possibility of the potential modulating effect of copper nanoparticles (Cu NPs) and polyinosinic-polycytidilic acid (poly I:C) on the binding of Cas9 to conserved omicron RNA (COR) and the resultant COR cleavage. This was also the first attempt to design a multi-component antiviral system, with each of the components having antiviral activity. To hypothesize the intracellular behavior of this approach, the characterization and understanding of the biophysical interactions among the different entities is necessary.

Biophysical characterization of the Cu NPs-poly I:C-COR-Cas9 system was performed using dynamic light scattering (DLS), two-dimensional fluorescence difference spectroscopy (2-D FDS), and agarose gel electrophoresis [30,31,32,33,34]. Shifts in the size, PDI, and 2-D FDS suggested biomolecular interactions between the protein and protein:RNA and copper nanoparticles. Whereas gel mobility shift is typically used to demonstrate these types of interactions, in this case, the change in patterns observed not only reflected this but also the progressive loss in staining intensity of the COR band suggests that the addition of the copper nanoparticles or the copper nanoparticles and poly I:C could facilitate cleavage. This greatly validates this study; the interaction of the Cas9 with the COR prevents some of the nucleic acid dye from binding, resulting in lowered stain intensity, but the addition of the nanoparticles and/or nanoparticles plus poly I:C both slightly alter the mobility as expected, and the loss in staining intensity is a reflection of the enhanced cleavage.

Cas9 is perhaps the best-studied and most well-characterized gene editing system. Beyond understanding its biophysical properties and biomolecular interactions with nanoparticles and RNA, the ability to direct its cleavage of RNA—for example, the highly conserved COR sequence in SARS-CoV-2—we believe has broad implications. In this case, poly I:C was used essentially as a proof-of-concept as a general RNA modifier, as inosine base-pairing to A, C or U would allow imperfect interaction of the I-containing strand, assuming some RNA exchange can occur to the Cas9–Cu complex. In the future, an antisense oligomer sequence with varying degrees of complementarity to the COR would likely yield more specific cleavage.

## Figures and Tables

**Figure 1 bioengineering-10-00512-f001:**
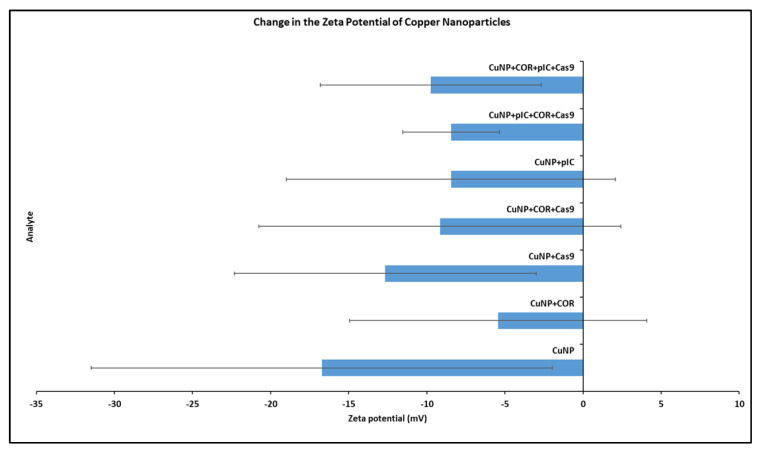
Change in the zeta potential of copper nanoparticles (Cu NPs) upon interaction with Cas9 and COR. The *x*-axis shows the zeta potential in millivolts (mV). The *y*-axis shows the different entities for which the zeta potential was analyzed. Order from bottom to top: (1) Cu NPs alone (2) Cu NPs + COR (3) Cu NPs + Cas9 (4) Cu NPs + COR + Cas9 (5) Cu NPs + poly I:C (6) Cu NPs + poly I:C + COR + Cas9 (7) Cu NPs + COR + poly I:C + Cas9. (Treatments 6 and 7 differ in the order of addition of poly I:C, COR, and Cas9). Error bars indicate the standard deviation.

**Figure 2 bioengineering-10-00512-f002:**
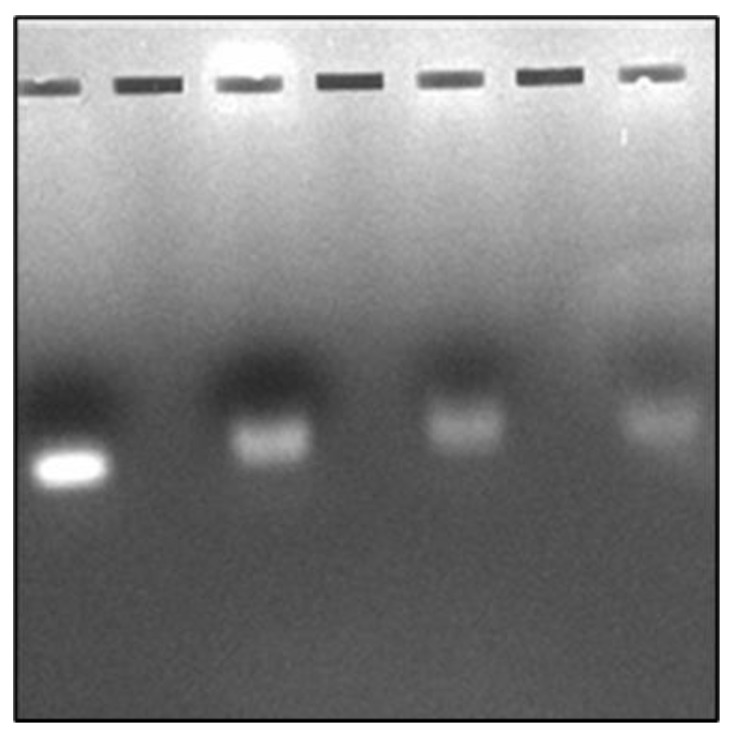
Interaction of COR with Cas9 in the presence of Cu NPs and/or poly I:C. Band (1) COR; Band (2) COR + Cas9; Band (3) COR + Cas9 + Cu NPs; Band (4) COR + Cas9 + Cu NPs + poly I:C as visualized by agarose gel electrophoresis (Bands counted from left to right).

**Figure 3 bioengineering-10-00512-f003:**
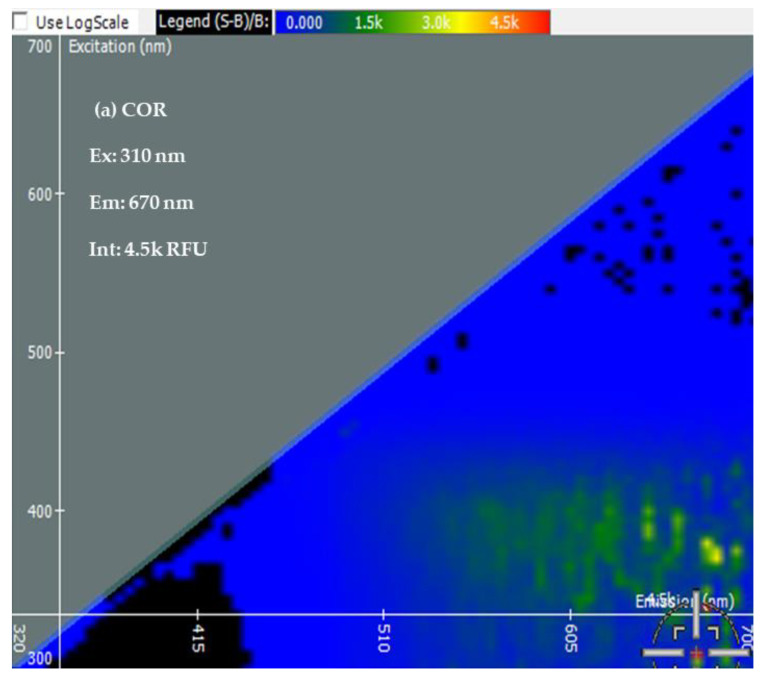
Spectral Signatures from Two-Dimensional Fluorescence Difference Spectroscopy (2-D FDS). Excitation and Emission spectra, and Fluorescence Intensities of (**a**) COR, (**b**) COR + Cu NPs, (**c**) COR + Cu NPs + Cas9, and (**d**) COR + Cu NPs + Cas9 + poly I:C (**e**) Comparison of spectral signatures (**a**–**d**).

**Table 1 bioengineering-10-00512-t001:** Sample treatments for dynamic light scattering (DLS) and zeta potential measurements. (Treatments 6 and 7 differ in the order of addition of poly I:C, COR, and Cas9.).

Sample Treatments	Contents (µL)			
	Cu NPs (1 mg/mL)	COR (1 µM)	Cas9 (0.04 µM)	poly I:C (1 mg/mL)
1. Cu NPs alone	20	-	-	-
2. Cu NPs + COR		1	-	-
3. Cu NPs + Cas9		-	14	-
4. Cu NPs + COR + Cas9		1	14	-
5. Cu NPs + poly I:C		-	-	1
6. Cu NPs + poly I:C + COR + Cas9		1	14	1
7. Cu NPs + COR + poly I:C + Cas9				

**Table 2 bioengineering-10-00512-t002:** Sample treatments for agarose gel electrophoresis.

Sample Treatments	Contents (µL)					
	Cu NPs	COR	Cas9	poly I:C	Milli-Q Water	Binding Buffer
1. COR	-	2	-	-	8	-
2. COR + Cas9	-	2	2	-	3	5
3. COR + Cas9 + Cu NPs	1	2	1	-	-	
4. COR + Cas9 + Cu NPs + poly I:C	1	2	1	1	-	

**Table 3 bioengineering-10-00512-t003:** Sample treatments for Two-Dimensional Fluorescence Difference Spectroscopy (2-D FDS).

Sample Treatments	Contents (µL)				
	Cu NPs	COR—Hoechst Dye Mix	Cas9	poly I:C	Milli-Q Water
1. COR	-	4	-	-	100
2. COR + Cu NPs	100		-	-	-
3. COR + Cu NPs + Cas9			25	-	75
4. COR + CuNP + Cas9 + poly I:C			25	10	75

**Table 4 bioengineering-10-00512-t004:** Size and polydispersity indices for copper nanoparticles (Cu NPs) interacting with Cas9, poly I:C, and COR from dynamic light scattering (DLS). Size is expressed as diameter values in nanometers (nm). Measurement values are expressed as values ± standard deviation.

Sample	Size (d. nm)	Polydispersity Index
1. Cu NPs	394 ± 97.8	0.49 ± 0.11
2. Cu NPs + COR	6005 ± 90.	0.62 ± 0.17
3. Cu NPs + Cas9	5513 ± 246	0.5 ± 0.04
4. Cu NPs + COR + Cas9	727 ± 2754	0.62 ± 0.19
5. Cu NPs + poly I:C	3149 ± 1468	0.43 ± 0.23
6. Cu NPs + poly I:C + COR + Cas9	4009 ± 2643	0.51 ± 0.16
7. Cu NPs + COR + poly I:C + Cas9	3738 ± 1979	0.47 ± 0.2

## Data Availability

Most of the data is contained within the article. Data not provided in the article can be found in the Appendix A.

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
