# Peer review of "Evidence of Copper Nanoparticles and Poly I:C Modulating Cas9 Interaction and Cleavage of COR (Conserved Omicron RNA)"

_bioengineering, 2023, doi:10.3390/bioengineering10050512_

Round 1

Reviewer 1 Report

This work describes the potential of copper nanoparticles and polyinosilic-polycytidilic acid on the binding of Cas9 to COR. The manuscript is well written and the experiments were planned and performed with due diligence. The Authors used the wide range of analytical and biological techniques including Dynamic Light Scattering, zeta potential measurement, agarose gel electrophoresis and fluorescence difference spectroscopy. The manuscript was written based on 50 very recent references describing exploration of novel methods of control of SARS-Cov-2 viral disease. The subject of the work undertaken by the Authors is very current due to the genetic mutations of the virus. In my opinion, the manuscript can be published in the present form in Bioengineering.

Author Response

Dear Reviewer,

Thank you for the positive comments recommending the manuscript to be published in Bioengineering in its present form.

Reviewer 2 Report

Karrer et al. reported an experimental study to explore the modulating effect of copper nanoparticles (Cu NP) and poly I:C for potential cleavage of conserved omicron RNA (COR) by Cas9 enzyme. The authors conducted several types of experiments to investigate the interaction of Cas9 and COR to Cu NP and showed COR’s degradation in the presence of Cas9, Cu NP, and poly I:C. This study might shed light on the future design of targeted COR cleavage pathways assisted by nanoparticles as new COVID-19 therapeutics.

The main goal and most of the contents of this manuscript are suitable for the scope of Bioengineering. However, the data presented are in general weak in statistical significance and lack proper interpretation and discussion. Therefore, I recommend reconsidering this manuscript after my major and minor concerns elaborated below are appropriately addressed.

General concept comments:

1.    In Figure 1 and Table 4, the error bars of almost all the data (Zeta potential, Size, Polydispersity Index) are so large that the statistical significance of the mean values is limited. More trials are needed for each experiment to improve the reliability and reproducibility of the data.

2.    Results 3.1 (even if it is assumed that the mean values of the data are statistically significant) lack in-depth discussion on the data from a chemical/physical perspective to reveal the mechanism of interaction between Cu NP and different component combinations. E.g.: Why are the Zeta potentials of CuNP+COR and CuNP+pIC higher than that of CuNP alone? How to interpret the different sizes and polydispersity indices as described on page 7, lines 217-225 – how does their increase/decrease indicate different interactions between Cu NP and other components? 

3.    Results 3.3, similar to 3.1, also lacks mechanistic interpretation of the fluorescence difference spectroscopy data. The authors should explain the meaning of excitation/emission peak wavelength and intensity, and elaborate on how would different spectral signatures “indicate the possibility of a crucial interaction among these species” (as claimed on page 10 line 308).  

Specific comments:

·  Page 7: “Changes in size and polydispersity indices were also monitored across the same 212 samples (Table 2)” should be (Table 4).

·  Figure 2: the difference of bands 2-4’s intensities are subtle to tell in this figure – more quantitative methods are needed for claims such as on page 8 lines 255-256: “However, the COR band is less intense then (band 3). The band intensity is further 255 reduced in the presence of poly I:C as evident with band 4”.

·  Table 5: These data would be much easier to interpret for the readers if shown as spectrum plots with peaks position and intensity labeled.

Author Response

Dear Reviewer,

Thank you, your insightful comments can help further improve the manuscript. Please see the attachment for responses to the comments. 

Reviewer 3 Report

I strongly suggest you to revise the whole part dealing with z-potential measurements. See, for instance, Figure 1 and Table 4. The latter is full of too many digits, as I have indicated. Normally, errors on such values are, as a rule, some percent.  The whole par must be critically revised. Other points are indicated in the enclosed file.  

Author Response

Dear Reviewer,

Thank you for the comments and suggestions. We have addressed the comments to the best of our knowledge. Please see the attachment for the responses.

Reviewer 4 Report

Dear authors,

The manuscript bioengineering-2182135-peer-review-v1, entitled ‘Evidence of Copper Nanoparticles and poly I:C modulating Cas9 interaction and cleavage of COR (Conserved Omicron RNA)’ presents the possibility of using poly I:C as a wobble guide RNA for Cas9 to bind to COR is explored.

Moreover, the authors suggest that their muti-component approach is expected to bring in the antiviral activities of each entity against SARS – CoV-2. For the definition in the intracellular behavior of this approach, the characterization and understanding of the biophysical interactions among the different entities is necessary.

Characterization methods used by the authors are as follows: Dynamic Light Scattering (DLS) and Zeta Potential Measurements, Agarose Gel Electrophoresis, and Two - Dimensional Fluorescence Difference Spectroscopy.

The overall manuscript text is written well, with good explanation and descriptions.

The conclusions are vast and should be compacted into main conclusions. Parts of the existing conclusions should be put in the discussion paragraph of the results.

However, taking into consideration the nature of the journal, and considering the possible soundness of this topic, the manuscript should be reconsidered in the way of bringing more experimental data and validation. The main idea is simple and interesting, so to prove such mechanisms more experimental data is needed.

I would suggest the authors to include more data on the Cu NP’s properties, how were manufactured, conditions of usage, special parameters varied for increasing the feasibility of their usage. A better ( proper) application explanation required for a broad reader target.

I propose this manuscript should be considered for publication in BIOENGINEERING after meeting these suggestions.

Recommendation: MAJOR Revision.

Author Response

Dear Reviewer,

Thank you, your insightful comments can help further improve the manuscript. Please see the attachment for the responses. 

Round 2

Reviewer 2 Report

In this revised version, all my previous concerns have been addressed. Therefore, I recommend its acceptance.

Author Response

Dear Reviewer,

Thank you for helping us to improve the manuscript and recommending its acceptance in Bioengineering.

Reviewer 3 Report

Correct Lines 44-46;

Line 159 there useless repetition

As to zeta-potential and DLS results, there are too many digits. I strongly suggest, for instance, -16.7 for zeta potential, and 394+/- 97 fro DLS. In either cases errors are reasonable. 

Correct Table 4

Lines 285-290: what do you mean?

Lines 325-329. Confuse.

Small corrections should be made. 

Author Response

Dear Reviewer,

Thank you for all the comments and for recommending the publication of our manuscript following minor revisions. Please see the attachment for our responses to your comments.

Reviewer 4 Report

Dear Authors,

The revised manuscript bioengineering-2182135-peer-review-v2, entitled ‘Evidence of Copper Nanoparticles and poly I:C modulating Cas9 interaction and cleavage of COR (Conserved Omicron RNA)’ presents the possibility of using poly I:C as a wobble guide RNA for Cas9 to bind to COR is explored.

The revised version has more info introduced in the manuscript body, as well as new evidences of the stated processes described in the text ( figures 3 on page 11).

The overall revised manuscript is improved but needs further attention:

-        Figure on page 8 between lines 218-219 needs to be deleted, since is doubling fig 1 on page 7;

-        In Figure 1, page 7, the text should be above blue horizontal columns, since it is hard to read in the form that it is now;

-        Imags in figure 3 are of low quality, should be increased the resolution, if possible;

-        Also a ‘clasical’ graph with excitation and fluorescence curves should be added in figure 3, in order to properly understand the following graphs in figure 3.

-        I would expect to find images (SEM or AFM) of the Copper nanoparticles (Cu NP; size 13-40 nm; dry powder).

After a quick and carefully check of the manuscript text and addressing the above issues, I would propose that the second revision version to be considered for publication in Bioengineering Journal:

Minor Revision

Author Response

(The authors gave the same response as above.)
